# Discrete-Valued Neural Networks Using Variational Inference

## Abstract

The increasing demand for neural networks (NNs) being employed on embedded devices has led to plenty of research investigating methods for training low precision NNs. While most methods involve a quantization step, we propose a principled Bayesian approach where we first infer a distribution over a discrete weight space from which we subsequently derive hardware-friendly low precision NNs. To this end, we introduce a probabilistic forward pass to approximate the intractable variational objective that allows us to optimize over discrete-valued weight distributions for NNs with sign activation functions. In our experiments, we show that our model achieves state of the art performance on several real world data sets. In addition, the resulting models exhibit a substantial amount of sparsity that can be utilized to further reduce the computational costs for inference.

## 1 Introduction

With the advent of deep neural networks (NNs) impressive performances have been achieved in many applications such as computer vision (Krizhevsky et al., 2012), speech recognition (Hinton et al., 2012), and machine translation (Sutskever et al., 2014), among others. However, the performance improvements are largely attributed to increasing hardware capabilities that enabled the training of ever-increasing network architectures. On the other side, there is also a growing interest in making NNs available for embedded devices with drastic memory and power limitations – a field with plenty of interesting applications that barely profit from the tendency towards larger and deeper network structures.

Thus, there is an emerging trend in developing NN architectures that allow fast and energy-efficient inference and require little storage for the parameters. In this paper, we focus on reduced precision methods that restrict the number of bits per weight while keeping the network structures at a decent size. While this reduces the memory footprint for the parameters accordingly, it can also result in drastic improvements in computation speed if appropriate representations for the weight values are used. This direction of research has been pushed towards NNs that require in the extreme case only a single bit per weight. In this case, assuming weights $w \in \{-1, 1\}$ and binary inputs $x \in \{-1, 1\}$, costly floating point multiplications can be replaced by cheap and hardware-friendly logical XNOR operations. However, training such NNs is inherently different as discrete valued NNs cannot be directly optimized using gradient based methods. Furthermore, NNs with binary weights exhibit their full computational benefits only in case the sign activation function is used whose derivative is zero almost everywhere, and, therefore, is not suitable for backpropagation.

Most methods for training reduced precision NNs either quantize the weights of pre-trained full precision NNs (Courbariaux et al., 2015a) or train reduced precision NNs by maintaining a set of full precision weights that are deterministically or stochastically quantized during forward or backward propagation. Gradient updates computed with the quantized weights are then applied to the full precision weights (Courbariaux et al., 2015b; Rastegari et al., 2016; Hubara et al., 2016). This approach alone fails if the sign activation function is used. A promising approach is based on the straight through gradient estimator (STE) (Bengio et al., 2013) which replaces the zero gradient of hard threshold functions by a non-zero surrogate derivative. This allows information in computation graphs to flow backwards such that parameters can be updated using gradient based optimization methods. Encouraging results are presented in (Hubara et al., 2016) where the STE is applied to the weight binarization and to the sign activation function. These methods, although showing

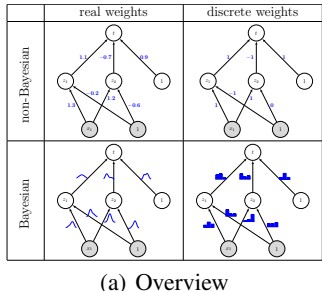
(a) Overview

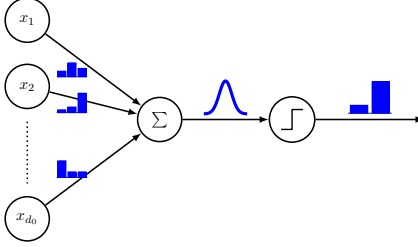
(b) Probabilistic forward pass

Figure 1: (a) Overview of real-valued vs. discrete-valued NNs and Bayesian vs. non-Bayesian NNs. The aim is to obtain a single discrete-valued NN (top right) with a good performance. We achieve this by training a distribution over discrete-valued NNs (bottom right) and subsequently deriving a single discrete-valued NN from that distribution. (b) Probabilistic forward pass: The idea is to propagate distributions through the network by approximating a sum over random variables by a Gaussian and subsequently propagating that Gaussian through the sign activation function.

convincing empirical performance, have in common that they appear rather heuristic and it is usually not clear whether they optimize any well defined objective.

Therefore, it is desired to develop principled methods that support discrete weights in NNs. In this paper, we propose a Bayesian approach where we first infer a distribution $q(\boldsymbol{W})$ over a discrete weight space from which we subsequently derive discrete-valued NNs. Thus, we can optimize over real-valued distribution parameters using gradient-based optimization instead of optimizing directly over the intractable combinatorial space of discrete weights. The distribution $q(\boldsymbol{W})$ can be seen as an exponentially large ensemble of NNs where each NN is weighted by its probability $q(\boldsymbol{W})$. Rather than having a single value for each connection of the NN, we now maintain a whole distribution for each connection (see bottom right of Figure 1(a)). To obtain $q(\boldsymbol{W})$, we employ variational inference where we approximate the true posterior $p(\boldsymbol{W}|\mathcal{D})$ by minimizing the variational objective $\mathrm{KL}(q(\boldsymbol{W})||p(\boldsymbol{W}|\mathcal{D}))$. Although the variational objective is intractable, this idea has recently received a lot of attention for real-valued NNs due to the reparameterization trick which expresses gradients of intractable expectations as expectations of tractable gradients (Rezende et al., 2014; Kingma & Welling, 2014; Blundell et al., 2015). This allows us to efficiently compute unbiased gradient samples of the intractable variational objective that can subsequently be used for stochastic optimization. Unfortunately, the reparameterization trick is only suitable for real-valued distributions which renders it unusable for our case. The recently proposed Gumbel softmax distribution (Jang et al., 2017; Maddison et al., 2017) overcomes this issue by relaxing one-hot encoded discrete distributions with probability vectors. Subsequently, the reparameterization trick can again be applied. However, for the sign activation function one still has to rely on the STE or similar heuristics. The log-derivative trick offers an alternative for discrete distributions to express gradients of expectations with expectations of gradients (Paisley et al., 2012). However, the resulting gradient samples are known to suffer from high variance. Therefore, the log-derivative trick is typically impractical unless suitable variance reduction techniques are used. This lack of practical methods has led to a limited amount of literature investigating Bayesian NNs with discrete weights (Soudry et al., 2014).

In this work, we approximate the intractable variational objective with a probabilistic forward pass (PFP) (Wang & Manning, 2013; Soudry et al., 2014; Hernandez-Lobato & Adams, 2015; Roth & Pernkopf, 2016). The idea is to propagate probabilities through the network by successively approximating the distributions of activations with a Gaussian and propagating this Gaussian through the sign activation function (Figure 1(b)). This results in a well-defined objective whose gradient with respect to the variational parameters can be computed analytically. This is true for discrete weight distributions as well as for the sign activation function with zero gradient almost everywhere. The method is very flexible in the sense that different weight distributions can be used in different layers. We utilize this flexibility to represent the weights in the first layer with 3 bits and we use ternary weights $w \in \{-1, 0, 1\}$ in the remaining layers.

In our experiments, we evaluate the performance of our model by reporting the error of (i) the most probable model of the approximate posterior $q(\boldsymbol{W})$ and (ii) approximated expected predictions

using the PFP. We show that averaging over small ensembles of NNs sampled from $\boldsymbol{W} \sim q(\boldsymbol{W})$ can improve the performance while inference using the ensemble is still cheaper than inference using a single full precision NN. Furthermore, our method exhibits a substantial amount of sparsity that further reduces the computational overhead. Compared to (Hubara et al., 2016), our method requires less precision for the first layer, and we do not introduce a computational overhead by using batch normalization which appears to be a crucial component of their method.

The paper is outlined as follows. In Section 2, we introduce the notation and formally define the PFP. Section 3 shows details of our model. Section 4 shows experiments. In Section 5 we discuss important issues concerning our model and Section 6 concludes the paper.

## 2    VARIATIONAL INFERENCE FOR DISCRETE-VALUED NEURAL NETWORKS

The structure of a feed-forward NN with $L$ layers is determined by the number of neurons $\{d_0, d_1, \ldots, d_L\}$ in each layer. Here, $d_0$ is the dimensionality of the input, $d_L$ is the dimensionality of the output, and $d_l$ for $0 < l < L$ is the number of hidden neurons in layer $l$. A NN defines a function $\boldsymbol{y} = \boldsymbol{x}^L = f(\boldsymbol{x}^0)$ by iteratively applying a linear transformation $\boldsymbol{a}^l = \boldsymbol{W}^l \boldsymbol{x}^{l-1}$ to the inputs from the previous layer followed by a non-linear function $\boldsymbol{x}^l = \boldsymbol{\phi}^l(\boldsymbol{a}^l)$. For $l < L$ we use the sign activation function $\phi^l(\boldsymbol{a}) = \mathbb{I}(\boldsymbol{a} \geq 0) - \mathbb{I}(\boldsymbol{a} < 0)$ which is applied element-wise to its inputs. For $l = L$, we use the softmax activation function $\mathrm{smax}_i(\boldsymbol{a}) = \exp(a_i) / \sum_j \exp(a_j)$. Note that the expensive softmax function does not need to be computed at test time.

NNs are parameterized by a set of weight matrices $\boldsymbol{W} = \{\boldsymbol{W}^l\}_{l=1}^L$ with $\boldsymbol{W}^l \in \mathbb{D}_l^{d_{l-1} \times d_l}$ where $\mathbb{D}_l$ is a finite set.[1] For a Bayesian treatment of NNs, we assume a prior distribution $p(\boldsymbol{W})$ over the discrete weights and interpret the output of the NN after the softmax activation as likelihood $p(\mathcal{D}|\boldsymbol{W})$ for the data set $\mathcal{D} = \{(\boldsymbol{x}_n^0, t_n)\}_{n=1}^N$. As the induced posterior distribution $p(\boldsymbol{W}|\mathcal{D}) \propto p(\mathcal{D}|\boldsymbol{W})p(\boldsymbol{W})$ is intractable for NNs of any decent size, we employ variational inference to approximate it by a simpler distribution $q(\boldsymbol{W}|\boldsymbol{\nu})$ by minimizing $\mathrm{KL}(q(\boldsymbol{W}|\boldsymbol{\nu})||p(\boldsymbol{W}|\mathcal{D}))$ with respect to the variational parameters $\boldsymbol{\nu}$. We adopt the common mean-field assumption where the approximate posterior factorizes into a product of factors $q(w|\nu_w)$ for each weight $w \in \boldsymbol{W}$.[2] The variational objective is usually transformed as

$$\mathrm{KL}(q(\boldsymbol{W})||p(\boldsymbol{W}|\mathcal{D})) = \mathrm{KL}(q(\boldsymbol{W})||p(\boldsymbol{W})) - \mathbb{E}_{q(\boldsymbol{W})}[\log p(\mathcal{D}|\boldsymbol{W})] + \log p(\mathcal{D}). \tag{1}$$

Minimizing this expression with respect to $\boldsymbol{\nu}$ does not involve the intractable posterior $p(\boldsymbol{W}|\mathcal{D})$ and the evidence $\log p(\mathcal{D})$ is constant with respect to the variational parameters $\boldsymbol{\nu}$. The KL term can be seen as a regularizer that pulls the approximate posterior $q(\boldsymbol{W})$ towards the prior distribution $p(\boldsymbol{W})$ whereas the expected log-likelihood captures the data. While the KL term is tractable if both the prior and the approximate posterior distribution assume independence of the weights, the expected log-likelihood is typically intractable due to a sum over exponentially many terms. We propose a PFP as closed-form approximation to the expected log-likelihood.

### 2.1    CLOSED-FORM APPROXIMATION OF THE EXPECTED LOG-LIKELIHOOD

The approximation of the expected log-likelihood resembles a PFP. In particular, we have

$$\mathbb{E}_{q(\boldsymbol{W})} \left[ \log p(t|\boldsymbol{x}^0, \boldsymbol{W}) \right] = \sum_{\boldsymbol{W}^1} \cdots \sum_{\boldsymbol{W}^L} q(\boldsymbol{W}) \log p(t|\boldsymbol{x}^0, \boldsymbol{W}) \tag{2}$$

$$\approx \sum_{\boldsymbol{W}^2} \cdots \sum_{\boldsymbol{W}^L} q(\boldsymbol{W}^{>1}) \int \mathcal{N}\left(\boldsymbol{a}^1 | \boldsymbol{\mu}_{a^1}, \boldsymbol{\sigma}_{a^1}\right) \log p(t|\boldsymbol{a}^1, \boldsymbol{W}^{>1}) d\boldsymbol{a}^1 \tag{3}$$

$$= \sum_{\boldsymbol{W}^2} \cdots \sum_{\boldsymbol{W}^L} \sum_{\boldsymbol{x}^1} q(\boldsymbol{W}^{>1}) \, \mathrm{Bernoulli}(\boldsymbol{x}^1 | \boldsymbol{\mu}_{x^1}) \log p(t|\boldsymbol{x}^1, \boldsymbol{W}^{>1}) \tag{4}$$

$$\vdots$$

$$\approx \int \mathcal{N}(\boldsymbol{a}^L | \boldsymbol{\mu}_{a^L}, \boldsymbol{\sigma}_{a^L}) \log \mathrm{smax}_t(\boldsymbol{a}^L) \, d\boldsymbol{a}^L, \tag{5}$$

---

[1] This notation allows us to have a different set $\mathbb{D}_l$, requiring a different number of bits, for each layer. For convenience, we include the biases in the weight matrices.

[2] In the sequel, we omit the dependency of the variational posterior $q(\boldsymbol{W})$ on the variational parameters $\boldsymbol{\nu}$.

where we defined $\boldsymbol{W}^{>k} = \{\boldsymbol{W}^l\}_{l=k+1}^{L}$. The overall aim is to successively get rid of the weights in each layer and consequently reduce the exponential number of terms to sum over. In the first approximation in (3), we approximate the activation distribution with Gaussians using a central limit argument. These Gaussian distributions are propagated through the sign activation function resulting in Bernoulli distributions in (4). These two steps are iterated until a Gaussian approximation of the output activations in (5) is obtained. This integral is approximated using a second-order Taylor expansion of the log-softmax around $\boldsymbol{\mu}_{a^L}$. In the following subsections we provide more details of the individual approximations.

### 2.1.1 APPROXIMATING THE ACTIVATIONS AND THE ACTIVATION FUNCTIONS

The activations of the neurons are computed as weighted sums over the outputs from the previous layers. Since the inputs and the weights are random variables, the activations are random variables as well. Given a sufficiently large number of input neurons, we can apply a central limit argument and approximate the activation distributions with Gaussians $\mathcal{N}(a_i^l|\mu_{a_i^l}, \sigma_{a_i^l})$. For computational convenience, we further assume that the activations within a layer are independent. Assuming that the inputs $x_j^{l-1}$ and the weights $w_{ij}^l$ are independent, we have

$$\mu_{a_i^l} = \sum_{j=1}^{d_{l-1}} \mu_{w_{ij}^l}\, \mu_{x_j^{l-1}}, \qquad \sigma_{a_i^l} = \sum_{j=1}^{d_{l-1}} \sigma_{w_{ij}^l}\, \mu_{(x_j^{l-1})^2} + \mu_{w_{ij}^l}^2 \left( \mu_{(x_j^{l-1})^2} - \mu_{x_j^{l-1}}^2 \right), \qquad (6)$$

where $\mu_{(x_j^{l-1})^2}$ denotes the raw second moment $\mathbb{E}[(x_j^{l-1})^2]$. In case of $l = 1$, we assume no variance at the inputs and thus the second term of $\sigma_{a_i^l}$ in (6) cancels.

In the next step, the Gaussian distributions $\mathcal{N}(a_i^l|\mu_{a_i^l}, \sigma_{a_i^l})$ over the activations are transformed by the sign activation function. The expectation of the resulting Bernoulli distribution with values $x \in \{-1, 1\}$ of the sign activation function is given by $\mu_{x_i^l} = \mathrm{erf}(\mu_{a_i^l}/(2\sigma_{a_i^l})^{\frac{1}{2}})$ where erf denotes the error function. The raw second moment as needed for (6) is $\mu_{(x_i^l)^2} = 1$.

### 2.1.2 EXPECTED LOG-LIKELIHOOD AT THE OUTPUTS

After iterating this approximation up to the last layer, it remains to calculate the expectation of the log-softmax with respect to the Gaussian approximation of the output activations in (5). Since this integral does not allow for an analytic solution, we approximate the log-softmax by its second-order Taylor approximation around the mean $\boldsymbol{\mu}_{a^L}$ with a diagonal covariance approximation, resulting in

$$\mathbb{E}_{\mathcal{N}(\boldsymbol{a}^L)} \left[ \log \mathrm{smax}_t(\boldsymbol{a}^L) \right] \approx \log \mathrm{smax}_t(\boldsymbol{\mu}_{a^L}) - \frac{1}{2} \sum_{t'} \sigma_{a_{t'}^L} \mathrm{smax}_{t'}(\boldsymbol{\mu}_{a^L}) \left( 1 - \mathrm{smax}_{t'}(\boldsymbol{\mu}_{a^L}) \right). \quad (7)$$

The maximization of the first term in (7) enforces the softmax output of the true class to be maximized, whereas the second term becomes relevant if there is no output close to one. For softmax outputs substantially different from zero or one, the product inside the sum is substantially larger than zero and the corresponding variance is penalized. In short, the second term penalizes large output variances if their corresponding output activation means are large and close to each other.

## 3 THE MODEL

In this section we provide details of the finite weight sets $\mathbb{D}_l$ and their corresponding prior and approximate posterior distributions, respectively. As reported in several papers (Hubara et al., 2016; Anderson & Berg, 2017), it appears to be crucial to represent the weights in the first layer using a higher precision. Therefore, we use three bits for the weights in the first layer and ternary weights in the remaining layers.

### 3.1 INPUT LAYER

We use $\mathbb{D}_1 = \{-0.75, -0.5, \ldots, 0.75\}$ for the first layer which can be represented as fixed point numbers using three bits. Note that $|\mathbb{D}_1| = 7$ and we could actually represent one additional value

with three bits. However, for the sake of symmetry around zero, we refrain from doing so as we do not want to introduce a bias towards positive or negative values. We empirically observed this range of values to perform well for our problems with inputs $x \in [-1, 1]$. The values in $\mathbb{D}_1$ can be scaled with an arbitrary factor at training time without affecting the output of the NN since only the sign of the activations is relevant. We investigated two different variational distributions $q(\boldsymbol{W}^1)$.

(i) General distribution: We store for each weight the seven logits corresponding to the unnormalized log-probabilities for each of the seven values. The normalized probabilities can be recovered using the softmax function. This simple and most general distribution for finite discrete distributions has the advantage that the model can express a tendency towards individual discrete values. Consequently, we expect the maximum of the distribution to be a reasonable value that the model explicitly favors. This is fundamentally different from training full precision networks and quantizing the weights afterwards. The disadvantage of this approach is that the number of variational parameters and the computation time for the means $\mu_w$ and variances $\sigma_w$ scales with the size of $\mathbb{D}_1$.

(ii) Discretized Gaussian: To get rid of the dependency of the number of parameters on $|\mathbb{D}_1|$, we also evaluated a discretized Gaussian. The distribution is parameterized by a mean $m_w$ and a variance $v_w$ and the logits of the resulting discrete distribution are given by $-(w - m_w)^2/(2\,v_w)$ for $w \in \mathbb{D}_1$ (Figure 2(a)). We denote this distribution as $\mathcal{N}_{\mathbb{D}_1}(m_w, v_w)$.[3] This parameterization has the advantage that only two parameters are sufficient to represent the resulting discrete distribution for an arbitrary size of $\mathbb{D}_1$. Furthermore, the resulting distribution is unimodal and neighboring values tend to have similar probabilities which appears natural. Nevertheless, there is no closed-form solution for the mean $\mu_w$ and variance $\sigma_w$ of this distribution and we have to compute a weighted sum involving the $|\mathbb{D}_1|$ normalized probabilities.

For the prior distribution $p(\boldsymbol{W}^1)$, we use for both aforementioned variational distributions the discretized Gaussian $\mathcal{N}_{\mathbb{D}_1}(0, \gamma)$ with $\gamma$ being a tunable hyperparameter. Computing the KL-divergence $\mathrm{KL}(q(\boldsymbol{W}^1)||p(\boldsymbol{W}^1))$ also requires computing a weighted sum over $|\mathbb{D}_1|$ values.

## 3.2 Hidden Layers

For the remaining layers we use ternary weights $w \in \mathbb{D}_l = \{-1, 0, 1\}$. We use a shifted binomial distribution, i.e. $w \sim \mathrm{Binomial}(2, w_p) - 1$. This distribution requires only a single parameter $w_p$ per weight for the variational distribution. The mean $\mu_w$ is given by $2w_p - 1$ and the variance $\sigma_w$ is given by $2w_p(1 - w_p)$. This makes the Bernoulli distribution an efficient choice for computing the required moments. It is convenient to select a binomial prior distribution $p(w) = \mathrm{Binomial}(2, 0.5)$ as it is centered at zero and we get $\mathrm{KL}(q(w)||p(w)) = |\mathbb{D}_l|(\log(2w_p)w_p + \log(2(1 - w_p))(1 - w_p))$.

These favorable properties of the binomial distribution might raise the question why it is not used in the first layer, especially since the required expectations, variances and KL-divergences are available in closed-form independent of the size of $\mathbb{D}_l$. We elaborate more on this in Section 5.

## 3.3 Activation Normalization

We normalize the activations of layer $l$ by $\sqrt{d_{l-1}}$.[4] This scales the activation means $\mu_a$ towards zero and keeps the activation variances $\sigma_a$ independent of the number of incoming neurons. Consequently, the expectation of the Bernoulli distribution after applying the sign activation function $\mu_x = \mathrm{erf}(\mu_a/(2\sigma_a)^{\frac{1}{2}})$ is less prone to be in the saturated region of the error function and, thus, gradients can flow backwards in the computation graph. Note that activation normalization influences only the PFP and does not change the classification result of individual NNs $\boldsymbol{W} \sim q(\boldsymbol{W})$ since only the sign of the activation is relevant.

## 3.4 Likelihood Weighting

The variational inference objective (1) does not allow to easily trade off between the importance of the expected log-likelihood $\mathbb{E}_{q(\boldsymbol{W})}[\log p(\mathcal{D}|\boldsymbol{W})]$ and the KL term $\mathrm{KL}(q(\boldsymbol{W})||p(\boldsymbol{W}))$. This is problematic since there are usually many more NN weights than there are data samples and the

---

[3]Note that the mean $\mu_w$ and the variance $\sigma_w$ of $\mathcal{N}_{\mathbb{D}_1}(m_w, v_w)$ are in general different from $m_w$ and $v_w$.
[4]The activation variance $\sigma_a$ is normalized by $d_{l-1}$.

KL term tends to be orders of magnitudes larger than the expected log-likelihood. As a result, the optimization procedure mainly focuses on keeping the approximate posterior $q(\boldsymbol{W})$ close to the prior $p(\boldsymbol{W})$ whereas the influence of the data is too small. We propose to counteract this issue by trading off between the expected log-likelihood and the KL term using a convex combination, i.e.,

$$-\lambda\, \mathbb{E}_{q(\boldsymbol{W})}[\log p(\mathcal{D}|\boldsymbol{W})] + (1-\lambda)\, \mathrm{KL}(q(\boldsymbol{W})||p(\boldsymbol{W})). \tag{8}$$

Here, $\lambda \in (0,1)$ is a tunable hyperparameter that can be interpreted as creating $\lambda/(1-\lambda)$ copies of the data set $\mathcal{D}$. A similar technique is used in (Mandt et al., 2016) to avoid getting stuck in poor local optima. Another approach to counteract this issue is the KL-reweighting scheme proposed in (Blundell et al., 2015). Due to the exponential weight decay, only a few minibatches are influenced by the KL term whereas the vast majority is mostly influenced by the expected log-likelihood.

## 4 EXPERIMENTS

We evaluated the performance of our model (NN VI) on MNIST (LeCun et al., 1998), variants of the MNIST data set (Larochelle et al., 2007), and a TIMIT data set (Zue et al., 1990). Details about the individual data sets can be found in the supplementary material. We selected a three layer structure with $d_1 = d_2 = 1200$ hidden units for all experiments. We evaluated both the *general* parameterization and the *Gaussian* parameterization as variational distribution $q(\boldsymbol{W}^{>1})$ for the first layer (Section 3.1), and a binomial variational distribution $q(\boldsymbol{W}^{>1})$ for the following layers (Section 3.2). We optimized the variational distribution using ADAM (Kingma & Ba, 2015) *without* KL reweighting (Blundell et al., 2015), and using rmsprop *with* KL reweighting, and report the experiment resulting in the better classification performance. For both optimization algorithms, we employ an exponential decay of the learning rate $\eta$ where we multiply $\eta$ after every epoch by a factor $\alpha \leq 1$. We use dropout (Srivastava et al., 2014) with dropout rate $p^{in}$ for the input layer and a common dropout rate $p^{hid}$ for the remaining hidden layers. We normalize the activation by $\sqrt{d_{l-1}\, p}$ where $p$ is either $p^{in}$ or $p^{hid}$ to consider dropout in the activation normalization. We tuned the hyperparameters $\lambda \in [0,1]$, $\gamma \in [10^{-5}, 10^0]$, $\eta \in [10^{-3}, 10^{-1}]$, $\alpha \in [0.975, 1]$, $p^{in} \in [0, 0.5]$, $p^{hid} \in [0, 0.8]$ with 50 iterations of Bayesian optimization (Snoek et al., 2012). We report the results for the *single* most probable model from the approximate posterior $\boldsymbol{W} = \arg\max_{\boldsymbol{W}} q(\boldsymbol{W})$. This model is indeed a low precision network that can efficiently be implemented in hardware. We also report results by computing predictions using the PFP which can be seen as approximating the expected prediction $\arg\max_t \mathbb{E}_{q(\boldsymbol{W})}[p(t|\boldsymbol{x}, \boldsymbol{W})]$ as it is desired in Bayesian inference.

We compare our model with real-valued NNs (NN real) trained with batch normalization (Ioffe & Szegedy, 2015), dropout, and ReLU activation function. We also evaluated the model from (Hubara et al., 2016) (NN STE) which uses batch normalization, dropout, real-valued weights in the first layer and binary weights $w \in \{-1, 1\}$ in the subsequent layers. The binary weights and the sign activation function are handled using the STE. For NN (real) and NN STE, we tuned the hyperparameters $\eta \in [10^{-4}, 10^{-2}]$, $\alpha \in [0.975, 1]$, $p^{in} \in [0, 0.5]$, and $p^{hid} \in [0, 0.8]$ on a separate held-out validation set using 50 iterations of Bayesian optimization.

The results are shown in Table 1. Our model (single) performs on par with NN (real) and it outperforms NN STE on the *TIMIT* data set and the more challenging variants of *MNIST* with different kinds of *background* artifacts. Interestingly, our model outperforms the other models on *MNIST Background* and *MNIST Background Random* by quite a large margin which could be due to the Bayesian nature of our model. The PFP outperforms the single most probable model. This is no surprise since the PFP uses all the available information from the approximate posterior $q(\boldsymbol{W})$. Overall, the performance of the *general* variational distribution seems to be slightly better than the discretized Gaussian at the cost of more computational overhead at training time. On a Nvidia GTX 1080 graphics card, a training epoch on the *MNIST* data set took approximately 8.8 seconds for NN VI general and 7.5 seconds for NN VI Gauss compared to 1.3 seconds for NN (real). The computational bottleneck of our method is the first layer since here the moments require computing weighted sums over all discrete values $w \in \mathbb{D}_1$.

### 4.1 ENSEMBLES OF DISCRETE NNS

Next, we approximate the expected predictions $\arg\max_t \mathbb{E}_{q(\boldsymbol{W})}[p(t|\boldsymbol{x}, \boldsymbol{W})]$ by sampling several models $\boldsymbol{W} \sim q(\boldsymbol{W})$. We demonstrate this on the *MNIST Rotated Background* data set. Figure 2(b)

Table 1: Classification errors [%] for different NN models. NN (real): Real-valued NNs with batch normalization, dropout and ReLU activation function. NN STE: NNs with batch normalization, dropout, sign activation function, real-valued weights in the first layer and binary weights in the remaining layers. NN VI (our method): 3 bits for the first layer and ternary weights for the remaining layers. We evaluated the *single* most probable model, the probabilistic forward pass (*pfp*), the *general* 3 bit distribution for the first layer, and the discretized *Gaussian* for the first layer.

| data set | NN (real) | NN STE | NN VI | | | |
|---|---|---|---|---|---|---|
| | | | general/pfp | general/single | Gauss/pfp | Gauss/single |
| MNIST | 1.280 | **1.240** | 1.300 | 1.430 | 1.290 | 1.320 |
| MNIST Basic | 2.878 | **2.614** | 2.736 | 2.806 | 2.656 | 3.064 |
| MNIST Background | 18.974 | 19.600 | **16.566** | 16.590 | 16.656 | 18.300 |
| MNIST Background Random | 12.008 | 11.808 | **9.430** | 9.716 | 9.574 | 10.158 |
| MNIST Rotated | **8.310** | 10.220 | 9.866 | 10.318 | 10.212 | 11.592 |
| MNIST Rotated Background | **48.894** | 52.586 | 49.030 | 52.768 | 49.700 | 51.780 |
| TIMIT | **18.957** | 21.939 | 20.441 | 20.857 | 19.429 | 21.051 |

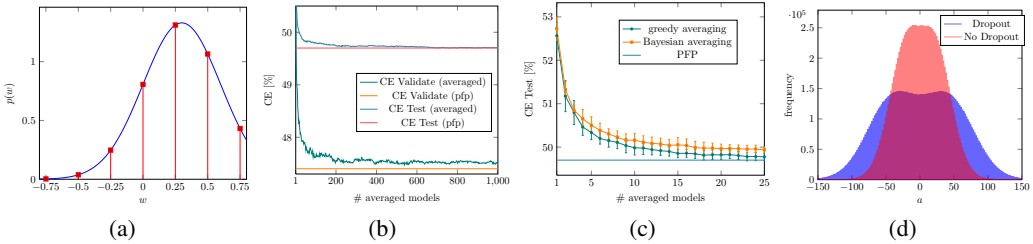

(a)        (b)        (c)        (d)

Figure 2: (a) Example discretized Gaussian with $m_w = 0.3$ and $\sqrt{v_w} = 0.3$. The red bars show the unnormalized probabilities of the corresponding weight values. (b) Classification errors (CE) of Bayesian averaging over several NN samples $\boldsymbol{W} \sim q(\boldsymbol{W})$. The CEs of the probabilistic forward pass (pfp) are shown as horizontal lines. (c) Test CE of Bayesian averaging using NN samples $\boldsymbol{W} \sim q(\boldsymbol{W})$ and averaging using the greedy forward selection strategy. (d) Histogram over activations of the same model once trained with dropout and once trained without dropout.

shows the classification error of Bayesian averaging over 1000 NNs sampled from the model with the best PFP performance using the discretized Gaussian. We see that the performance approaches the PFP which indicates that the PFP is a good approximation to the true expected predictions. However, the size of the ensemble needed to approach the PFP is quite large and the computation time of evaluating a large ensemble is much larger than a single PFP. Therefore, we investigated a greedy forward selection strategy, where we sample 100 NNs out of which we include only the NN in the ensemble which leads to the lowest error. This is shown in Figure 2(c). Using this strategy results in a slightly better performance than Bayesian averaging. Most importantly, averaging only a few models results in a decent performance increase while still allowing for faster inference than full precision NNs.

## 4.2 COMPUTATIONAL COSTS AND SPARSITY

Our NNs obtained by taking the most probable model from $q(\boldsymbol{W})$ can be efficiently implemented in hardware. They require only multiplications with 3 bit fixed point values as opposed to multiplications with floating point values in NN (real) and NN STE. In the special case of image data, the inputs are also given as 8 bit fixed point numbers. By scaling the inputs and the weights from the first layer appropriately, this results in an ordinary integer multiplication while leaving the output of the sign activation function unchanged. In the following layers we only have to compute multiplications as logical XNOR operations and accumulate -1 and +1 values for the activations.

Table 2 shows the fraction of non-zero weights of the best performing single models from Table 1. Especially in the input layer where we have our most costly 3 bit weights, there are a lot of zero weights on most data sets. This can be utilized to further reduce the computational costs. For example, on the *MNIST Background Random* data set, evaluating a single NN requires only approximately 23000 integer multiplications and 1434000 XNOR operations instead of approximately 2393000 floating point multiplications.

Table 2: The fraction [%] of non-zero weights of the best performing single model for individual layers and for all layers combined.

| data set | Layer 1 | Layer 2 | Layer 3 | Overall |
|---|---|---|---|---|
| MNIST | 22.49 | 66.26 | 95.92 | 49.20 |
| MNIST Basic | 36.78 | 96.40 | 99.84 | 72.97 |
| MNIST Background | 0.43 | 96.91 | 99.73 | 58.99 |
| MNIST Background Random | 2.44 | 98.72 | 99.99 | 60.88 |
| MNIST Rotated | 81.86 | 93.12 | 99.82 | 88.72 |
| MNIST Rotated Background | 4.92 | 83.98 | 88.93 | 52.92 |
| TIMIT | 61.42 | 97.80 | 99.98 | 95.35 |

## 5 DISCUSSION

The presented model has many tunable parameters, especially the type of variational distributions for the individual layers, that heavily influence the behavior in terms of convergence at training time and performance at test time. The binomial distribution appears to be a natural choice for evenly spaced values with many desirable properties. It is fully specified by only a single parameter, and its mean, variance, and KL divergence with another binomial has nice analytic expressions. Furthermore, neighboring values have similar probabilities which rules out odd cases in which, for instance, there is a value with low probability in between of two values with high probability.

Unfortunately, the binomial distribution is not suited for the first layer as here it is crucial to be able to set weights with high confidence to zero. However, when favoring zero weights by setting $w_p = 0.5$, the variance of the binomial distribution takes on its largest possible value. This might not be a problem in case predictions are computed as the true expectations with respect to $q(\boldsymbol{W})$ as in the PFP, but it results in bad classification errors when deriving a single model from $q(\boldsymbol{W})$. We also observed that using the binomial distribution in deeper layers favor the weights $-1$ and $1$ over $0$ (cf. Table 2). This might indicate that binary weights $w \in \{-1, 1\}$ using a Bernoulli distribution could be sufficient, but in our experiments we observed this to perform worse. We believe this to stem partly from the importance of the zero weight and partly from the larger variance of $4w_p(1 - w_p)$ of the Bernoulli distribution compared to the variance of $2w_p(1 - w_p)$ of the binomial distribution.

Furthermore, there is a general issue with the sign activation functions if the activations are close to zero. In this case, a small change to the inputs can cause the corresponding neuron to take on a completely different value which might have a large impact on the following layers of the NN. We found dropout to be a very helpful tool to counteract this issue. Figure 2(d) shows histograms of the activations of the second hidden layer for both a model trained with dropout and the same model trained without dropout. We can see that without dropout the activations are much closer to zero whereas dropout introduces a much larger spread of the activations and even causes the histogram to decrease slightly in the vicinity of zero. Thus, the activations are much more often in regions that are stable with respect to changes of their inputs which makes them more robust. We believe that such regularization techniques are crucial if the sign activation function is used.

## 6 CONCLUSION

We introduced a method to infer NNs with low precision weights. As opposed to existing methods, our model neither quantizes the weights of existing full precision NNs nor does it rely on heuristics to compute "approximated" gradients of functions whose gradient is zero almost everywhere. We perform variational inference to obtain a distribution over a discrete weight space from which we subsequently derive a single discrete-valued NN or a small ensemble of discrete-valued NNs. Our method propagates probabilities through the network which results in a well defined function that allows us to optimize the discrete distribution even for the sign activation function. The weights in the first layer are modeled using fixed point values with 3 bits precision and the weights in the remaining layers have values $w \in \{-1, 0, 1\}$. This reduces costly floating point multiplications to cheaper multiplications with fixed point values of 3 bits precision in the first layer, and logical XNOR operations in the following layers. In general, our approach allows flexible bit-widths for each individual layer. We have shown that the performance of our model is on par with state of the art methods that use a higher precision for the weights. Furthermore, our model exhibits a large amount of sparsity that can be utilized to further reduce the computational overhead.

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

## A  DATA SETS

### A.1  MNIST

The MNIST data set (LeCun et al., 1998) contains grayscale images of size $28 \times 28$ showing hand-written digits. It is split into 50000 training samples, 10000 validation samples, and 10000 test samples. The task is to classify the images to digits. The pixel intensities are normalized to the range $[-1, 1]$ by dividing through 128 and subtracting 1. We use the MNIST data set in the *permutation-invariant* setting where the model is not allowed to use prior knowledge about the image structure, i.e., convolutional NNs are not allowed.

### A.2  VARIANTS OF MNIST

The variants of the MNIST data set (Larochelle et al., 2007) contain images of size $28 \times 28$ showing images of the original MNIST data set that have been transformed by various operations in order to obtain more challenging data sets. The variants of the MNIST data set are split into 10000 training samples, 2000 validation samples and 50000 test samples. In particular, there are the following variants:

- *MNIST Basic*: This data set has not been transformed. The data set is merely split differently into training, validation, and test set, respectively.
- *MNIST Background*: The background pixels of the images have been replaced by random image patches.
- *MNIST Background Random*: The background pixels of the images have been set to a uniformly random pixel value.
- *MNIST Rotated*: The images are randomly rotated.
- *MNIST Rotated Background*: The transformations from *MNIST Rotated* and *MNIST Background* are combined

Some samples of the individual data sets are shown in Figure 3. We also normalized the pixel intensities of these data sets to lie in the range $[-1, 1]$.

### A.3  TIMIT

The TIMIT data set (Zue et al., 1990) contains samples of 92 features representing a phonetic segment. The task is to classify the phonetic segment to one of 39 phonemes. The data is split into 140173 training samples, 50735 validation samples (test) and 7211 test samples (core test). Details on data preprocessing can be found in (Halberstadt & Glass, 1997). We normalized the features to have zero mean and unit variance.

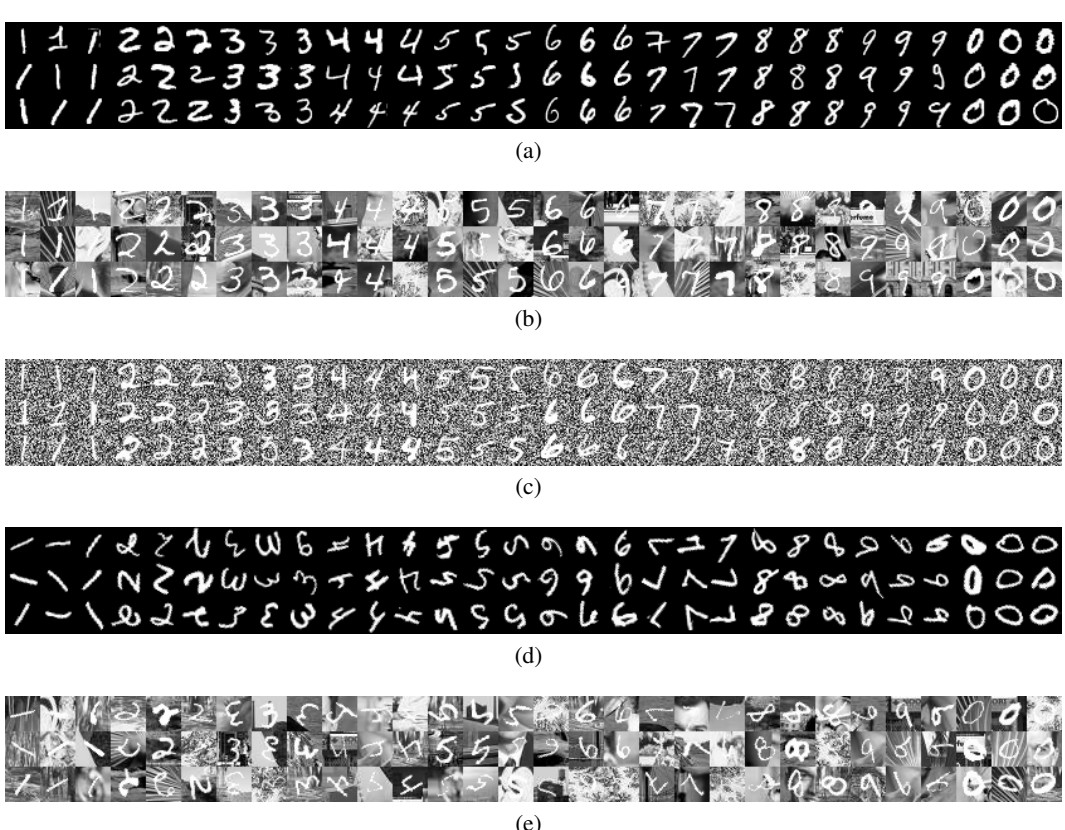

Figure 3: Some samples from the *MNIST* data set and variants of the *MNIST* data set. (a) *MNIST* and *MNIST Basic*, (b) *MNIST Background*, (c) *MNIST Background Random*, (d) *MNIST Rotated*, and (e) *MNIST Rotated Background*.

