# OpenReview forum: "Discrete-Valued Neural Networks Using Variational Inference"
_ICLR.cc/2018/Conference — Reject_

### Official Review · AnonReviewer2 · 2017-11-19
**This is outside of my areas of expertise. This is an educated guess. Marginal accept.**

**Rating:** 6
**Confidence:** 1

**Review:**

Summary:
The paper considers a Bayesian approach in order to infer the distribution over a discrete weight space, from which they derive hardware-friendly low precision NNs. This is an alternative to a standard quantization step, often performed in cases such as emplying NNs on embedded devices.
The NN setting considered here contains sign activation functions.
The experiments conducted show that the proposed model achieves nice performance on several real world data Comments

Due to an error in the openreview platform, I didn't have the chance to bid on time. This is not within my areas of expertise. Sorry for any inconvenience.

---

### Official Review · AnonReviewer1 · 2017-11-28
**New approach to train ternary-weight NNs, unclear advantages with respect to previous works**

**Rating:** 5
**Confidence:** 4

**Review:**

In this work, discrete-weight NNs are trained using the variational Bayesian framework, achieving similar results to other state-of-the-art models. Weights use 3 bits on the first layer and are ternary on the remaining layers.


- Pros:

The paper is well-written and connections with the literature properly established.

The approach to training discrete-weights NNs, which is variational inference, is more principled than previous works (but see below).

- Cons:

The authors depart from the original motivation when the central limit theorem is invoked. Once we approximate the activations with Gaussians, do we have any guarantee that the new approximate lower bound is actually a lower bound? This is not discussed. If it is not a lower bound, what is the rationale behind maximizing it? This seems to place this work very close to previous works, and not in the "more principled" regime the authors claim to seek.

The likelihood weighting seems hacky. The authors claim "there are usually many more NN weights than there are data samples". If that is the case, then it seems that the prior dominating is indeed the desired outcome. A different, more flat prior (or parameter sharing), can be used, but the described reweighting seems to be actually breaking a good property of Bayesian inference, which is defecting to the prior when evidence is lacking.

In terms of performance (Table 1), the proposed method seems to be on par with existing ones. It is unclear then what the advantage of this proposal is.

Sparsity figures are provided for the current approach, but those are not contrasted with existing approaches. Speedup is claimed with respect to an NN with real weights, but not with respect existing NNs with binary weights, which is the appropriate baseline.


- Minor comments:

Page 3: Subscript t and variable t is used for the targets, but I can't find where it is defined.

Only the names of the datasets used in the experiments are given, but they are not described, or even better, shown in pictures (maybe in a supplementary).

The title of the paper says "discrete-valued NNs". The weights are discrete, but the activations and outputs are continuous, so I find it confusing. As a contrast, I would be less surprised to hear a sigmoid belief network called a "discrete-valued NN", even though its weights are continuous.

---

> ### Author Response · Authors · 2017-12-07
> **Response to AnonReviewer1**
>
> Thank you for your valuable comments.
>
> - A general comment:
> It appears that this review is mainly concerned with our method being slightly off from the textbook Bayesian approach. One major motivation of obtaining discrete-valued NNs by first inferring a distribution is that the distribution parameters are real-valued and can therefore be optimized with gradient based optimization. Directly optimizing the discrete weights would result in an intractable combinatorial optimization problem.
>
> - Issue with Gaussian approximation:
> We agree that, due to the Gaussian approximation, we are not maximizing a lower bound anymore (at least we did not investigate on this). However, our motivation was to come up with a principled scheme to obtain resource-efficient NNs with discrete weights that achieve good performance, and, therefore, we accept to slightly depart from the full Bayesian path. By "more principled" we refer to existing work on resource efficient NNs (rather than to work in the Bayesian literature) where mostly some quantization step is applied or, in the case of the straight through estimator, a gradient that is clearly zero is "approximated" by something non-zero. With these methods, it is often not even clear if the gradient update procedures are optimizing any objective. We believe that this direction of research requires more principled methods such as the presented one.
>
> - Likelihood weighting:
> We believe that the prior-term dominating the likelihood-term is an artifact of variational inference by minimizing the KL-divergence between approximate and true posterior that especially manifests itself in case of NNs. In many hierarchical Bayesian models, there is a latent variable per data point that one aims to estimate and therefore the numbers of KL-terms and likelihood-terms are balanced at all times. For NNs, the number of KL-terms is fixed as soon as we fix the structure of the NN, and, as is commonly known, larger NNs tend to perform better. Hence, using vanilla KL-minimization results in a dilemma if we want to estimate the parameters of a NN whose number of parameters is orders of magnitudes larger than the number of data samples. Using a flat (constant) prior only partly solves this problem as an entropy-term, which itself dominates the likelihood-term, would still be present. This entropy-term would cause the approximate posterior to take on larger variances which would again severely degrade performance. We agree that parameter sharing could help since it would reduce the number of KL-terms, but this would result in a different model.
>
> - Performance:
> Our "single" model outperforms the NN STE [1] by ~2-3% on MNIST background and MNIST background random, respectively. On the other data sets we are on par. Furthermore, we achieve similar performance as NN (real) which is more computationally expensive to evaluate.
>
> - Advantages of our model (compared to other resource-efficient methods):
>  - Well defined objective function
>  - Probabilistic forward pass simultaneously handles both discrete distributions and the sign activation function
>  - Flexible choice of discrete weight space; can be different in each layer (other methods are often very rigid in this regard)
>  - Low precision weights in the first layer
>  - Additional information available in the approximate posterior
>
> - Sparsity:
> Regarding other methods: Binary and real weights (e.g. as in [1]), respectively, do not exhibit any sparsity at all, i.e. each connection of the NN is present. We point out that our method introduces, at least on some data sets, a substantial amount of sparsity that can be utilized to reduce computational costs. This was not a design goal in itself and we do not claim that our method is competitive with other approaches that explicitly aim to achieve sparsity. We think that the way that sparsity arises in our model is compelling: The value zero is explicitly modeled and we do not prune weights after training by some means of post-processing.
>
> - Minor comment on the title:
> It seems there is a misunderstanding. In our experiments, the "single" model refers to a single low-resource NN obtained as the most probable NN from the approximate posterior. In this NN, the activations and outputs are *not* continuous - given that the inputs are low-precision fixed-point values (as in images), the activations in the first hidden layer are obtained by low-precision fixed point operations (or equivalently integer operations), and the activations in the following layers are obtained by accumulating -1 and +1. The activation functions are sign functions that result in either -1 or +1. The output activations are also integer valued as they only accumulate -1 and +1 (the softmax is not needed at test time). Only for the "pfp" model and during optimization we have to deal with real-valued quantities.
>
> - Other minor comments:
> Thank you, we will use your comments to improve the paper.
>
> [1] Hubara et al., Binarized neural networks, NIPS 2016

---

### Official Review · AnonReviewer3 · 2017-11-28

**Rating:** 5
**Confidence:** 4

**Review:**

The authors consider the problem of ultra-low precision neural networks motivated by
limited computation and bandwidth. Their approach first posits a Bayesian neural network
a discrete prior on the weights followed by central limit approximations to efficiently
approximate the likelihood. The authors propose several tricks like normalization and cost
rescaling to help performance. They compare their results on several versions of MNIST. The
paper is promising, but I have several questions:

1) One major concern is that the experimental results are only on MNIST. It's important
to have another (larger) dataset to understand how sensitive the approach is to
characteristics of the data. It seems plausible that a more difficulty problem may
require more precision.

2) Likelihood weighting is related to annealing and variational tempering

3) The structure of the paper could be improved:
 - The introduction contains way too many details about the method
    and related work without a clear boundary.
 - I would add the model up front at the start of section 2
 - Section 2.1 could be reversed or equations 2-5 could be broken with text
   explaining each choice

4) What does training time look like? Is the Bayesian optimization necessary?

---

> ### Author Response · Authors · 2017-12-07
> **Response to AnonReviewer3**
>
> Thank you for your valuable comments.
>
> - Data sets:
> We are currently running another experiment on a TIMIT data set (phoneme classification) which is larger (N~140k), has more classes (39), but has less features (92). Other papers on resource efficient NNs typically evaluate on larger image tasks like CIFAR-10, SVHN and ImageNet. However, we refrain from doing so as we have not yet considered convolutional NNs and it is known that plain fully-connected NNs are far too weak to come close to state-of-the-art performance.
>
> - Likelihood-weighting and annealing/variational tempering:
> We assume you are referring to [1]. We agree that our weighting scheme is similar to these methods but they are used in different ways and for different purposes. We will comment on this in our revision.
>
> - Structure of the paper:
> Thank you for pointing this out. We will consider this in our revision.
>
> - Training time and Bayesian optimization:
> Training time naturally increases compared to the training time of plain NNs since computing the required first and second moments of the probabilistic forward pass is more time-consuming. On a Nvidia GTX 1080 graphics card, a training epoch on MNIST with a minibatch size of 100 (500 parameter updates) takes approximately 8.8 seconds for the general 3 bit distribution and 7.5 seconds for the discretized Gaussian distribution compared to 1.3 seconds for plain NNs. Especially the first layer is a bottleneck since here the moments require computing weighted sums over all discrete values. Of course we could have hand-tuned the hyperparameters, but we believe that Bayesian optimization is a useful tool that relieves us from putting too much effort into finding suitable hyperparameters. Furthermore, it allows for a fair comparison between models by evaluating them for the same number of iterations. We will include the training times in our revision.
>
> [1] S. Mandt et al., Variational Tempering, AISTATS 2016

---

### Author Response · Authors · 2017-12-18
**Revision**

We added a revision of our paper where we changed the following aspects.

(1) We left the structure of the paper largely unchanged. We removed some details about our method from the introduction but we kept the related work there as it is needed to motivate our work and to highlight the gap in the literature we intend to fill.

(2) We added results of experiments performed on a larger TIMIT data set for phoneme classification. Furthermore, the data sets are now described in more detail in the supplementary material. Our model (single: with sign activation function, 3 bit weights in the input layer, and ternary weights in the following layers) performs on par with NN (real) and it outperforms NN STE on the TIMIT data set and the more challenging variants of MNIST with different kinds of background artifacts.

(3) We added the training time in the experiments section.

(4) A few minor changes.

---

### Decision · Program_Chairs · 2018-01-29
**ICLR 2018 Conference Acceptance Decision**

**Decision:**

Reject

**Comment:**

This paper presents a somewhat new approach to training neural nets with ternary or low-precision weights.  However the Bayesian motivation doesn't translate into an elegant and self-tuning method, and ends up seeming kind of complicated and ad-hoc.  The results also seem somewhat toy.  The paper is fairly clearly written, however.